# Impact of Sertraline, Fluoxetine, and Escitalopram on Psychological Distress among United States Adult Outpatients with a Major Depressive Disorder

**DOI:** 10.3390/healthcare11050740

**Published:** 2023-03-03

**Authors:** Kwame Adjei, Georges Adunlin, Askal Ayalew Ali

**Affiliations:** 1Department of Economic, Social and Administrative Pharmacy, Florida Agricultural and Mechanical University, Tallahassee, FL 32307, USA; 2Department of Pharmaceutical, Social and Administrative Sciences, Samford University, Birmingham, AL 35229, USA

**Keywords:** Sertraline, Fluoxetine, Escitalopram, psychological distress, monotherapy, Kessler Index, major depressive disorder, serotonin transporter, antidepressants

## Abstract

How impactful is the use of Sertraline, Fluoxetine, and Escitalopram monotherapy on psychological distress among adults with depression in the real world? Selective serotonin reuptake inhibitors (SSRIs) are the most commonly prescribed antidepressants. Medical Expenditure Panel Survey (MEPS) longitudinal data files from 1 January 2012 to 31 December 2019 (panel 17–23) were used to assess the effects of Sertraline, Fluoxetine and Escitalopram on psychological distress among adult outpatients diagnosed with a major depressive disorder. Participants aged 20–80 years without comorbidities, who initiated antidepressants only at rounds 2 and 3 of each panel, were included. The impact of the medicines on psychological distress was assessed using changes in Kessler Index (K6) scores, which were measured only in rounds 2 and 4 of each panel. Multinomial logistic regression was conducted using the changes in the K6 scores as a dependent variable. A total of 589 participants were included in the study. Overall, 90.79% of the study participants on monotherapy antidepressants reported improved levels of psychological distress. Fluoxetine had the highest improvement rate of 91.87%, followed by Escitalopram (90.38%) and Sertraline (90.27%). The findings on the comparative effectiveness of the three medications were statistically insignificant. Sertraline, Fluoxetine, and Escitalopram were shown to be effective among adult patients suffering from major depressive disorders without comorbid conditions.

## 1. Introduction

Approximately 15 million physician office visits with depressive disorders as the primary diagnosis were recorded in 2019 [1]. An estimated 21 million adults and 4.1 million adolescents aged 12 to 17 in the USA in 2017 had at least one major depressive episode, representing 8.4% and 17% of the USA population, respectively [2]. According to the World Health Organization, depression is ranked as the most significant cause of disability worldwide and contributes heavily to the global disease burden [3]. Depression is the major contributing factor to suicide and ischemic heart disease [4].

“According to the Global Burden of Disease study, major depressive disorder was recorded as the mental health disorder with the highest economic burden accounting for 2.7 million disability-adjusted life years in 2016” [5]. In 2018, the economic burden of depression was estimated at USD 326 billion, representing an increase of 37.9% between 2010 and 2018 [6]. “The Center for Disease Control emphasizes that over the past two decades, the use of antidepressants has experienced tremendous growth, making them one of the most expensive and third most prescribed drugs in the USA” [7].

First-generation antidepressants, such as tricyclic antidepressants and monoamine oxidase inhibitors, used to be the main treatment for depression, but they are no longer preferred in many clinics due to their serious side effects, such as orthostatic hypotension and insomnia [8,9,10]. Second-generation antidepressants, including selective serotonin reuptake inhibitors (SSRIs), serotonin and norepinephrine reuptake inhibitors (SNRIs), and dopamine reuptake inhibitors, have fewer side effects than first-generation antidepressants [11]. Fluoxetine and Sertraline were among the first SSRIs approved for depression treatment in the 1990s, and Escitalopram was introduced in 2003 [12]. Although the different classes of second-generation antidepressants have similar effectiveness on quality of life, they differ in their pharmacokinetics, pharmacodynamics, and side effects, which may impact treatment selection [13]. Fluoxetine has a lower specificity of serotonin transporter (SERT) than other SSRIs, but a better binding specificity than tricyclic antidepressants and monoamine oxidase inhibitors [14,15]. Fluoxetine can lead to weight loss, agitation, and anxiety; Sertraline is associated with a higher incidence of diarrhea; and Escitalopram has a higher likelihood than other SSRIs of causing QT prolongation [16,17,18].

In clinical practice, second-generation antidepressants are prescribed for many conditions other than depression, such as anxiety, sleeping disorders, psychosis, and neuropathic pain [19]. “Sertraline is currently approved for major depressive disorder, obsessive-compulsive disorder, panic disorder, post-traumatic stress disorder, seasonal affective disorder, and premenstrual dysphoric disorder” [14]. Escitalopram is also used in the management of generalized anxiety disorder, while Fluoxetine is used in the treatment of premenstrual dysphoric disorder [14]. The choice of antidepressants is influenced by drug profiles, physician characteristics, patient characteristics, and other factors such as comorbidities [20,21].

“Psychological distress refers to non-specific symptoms of stress, anxiety, and depression. High levels of psychological distress are indicative of impaired mental health and may reflect common mental disorders, like depressive and anxiety disorders” [22]. Research has shown that individuals with depression often experience high levels of psychological distress in various areas of life, which leads to a decline in physical, emotional, and social functioning [23]. Physical symptoms of depression, such as fatigue and changes in appetite and sleep patterns, can negatively impact an individual’s ability to engage in physical activity and maintain good physical health [23,24]. Emotional symptoms, such as feelings of sadness and hopelessness, can lead to difficulty in maintaining personal relationships and a lack of interest in activities. Social functioning may also be affected, as individuals with depression may withdraw from social interactions and have difficulty in forming and maintaining relationships [23].

In addition to the negative impact of psychological distress, depression also increases the risk of various physical health problems, such as cardiovascular disease, diabetes, and obesity which can be attributed to unhealthy coping mechanisms such as overeating, lack of physical activity, and substance abuse [24,25]. It is important for individuals with depression to receive appropriate treatment and support to improve their overall well-being and functioning.

There are widely used survey instruments for measuring psychological distress in people with depression, such as the Patient Health Questionnaire-9 (PHQ-9), the Beck Depression Inventory (BDI), and the Kessler Psychological Distress Scale (K6). “The PHQ-9 is a self-administered questionnaire that assesses the presence and severity of depressive symptoms over the past two weeks, consisting of nine items rated on a four-point Likert scale” [26]. The BDI is a 21-item self-report inventory that measures the presence and severity of depression symptoms over the past two weeks, assessing symptoms such as sadness, hopelessness, and self-esteem, each rated on a four-point Likert scale [27]. The K6 is a brief, self-administered questionnaire that assesses symptoms of non-specific psychological distress over the past 30 days, consisting of six items rated on a five-point Likert scale. A score of 13 or higher on the K6 is considered to indicate clinically significant psychological distress [28].

The K6 is a reliable and valid measure of psychological distress among patients with depression. It has good test–retest reliability, with a correlation coefficient of 0.8, and strong concurrent validity, as it correlates well with other measures of depression and anxiety and is able to discriminate between patients with depression and those without depression [28,29].

Over 40% of depression patients fail to improve with conventional treatment, which involves using a single antidepressant agent at a prescribed dose and duration [30,31,32,33]. In spite of the considerable amount of data available on the clinical efficacy of second-generation antidepressants, there remains insufficient evidence on the real-world impact of the most widely prescribed second-generation antidepressants on patient-reported outcomes.

This study evaluated the effectiveness of the most commonly prescribed antidepressants, Sertraline, Fluoxetine and Escitalopram, on psychological distress among various subgroup populations based on age, race, and sex using a nationally representative sample in the United States.

## 2. Materials and Methods

### 2.1. Data Source

The current retrospective longitudinal study was conducted to examine the effectiveness of Sertraline, Fluoxetine, and Escitalopram monotherapy on psychological distress as a patient-reported outcome among the non-institutionalized US population using the Medical Expenditure Panel Survey (MEPS). The MEPS data used in this study spanned the period 1 January 2012 to 31 December 2019 (panel 17 to panel 23) [34].

The MEPS is a nationally representative estimate of health care use, expenditure, sources of payment, health insurance coverage, and demographic characteristics, additionally providing data on respondents’ health status, employment, access to care, and satisfaction with healthcare [34]. “The National Health Interview Survey (NHIS) uses a stratified, multistage probability cluster sampling design which provides a nationally representative sample of the U.S. civilian, non-institutionalized population” [34]. “A computer assisted personal interviewing (CAPI) technology is used to collect information about each household member and the information collected for a sampled household is reported by a single household respondent. Verification of patient’s reports are conducted through a survey response from their healthcare providers and contacting the pharmacies where the participants reported of filling their prescribed medicines” [34,35]. The panel design of the survey comprises five rounds of interviews covering two full calendar years (Figure 1).

Depression was defined as a major depressive episode that affects mood, behavior, and overall health, causing prolonged feelings of sadness, emptiness, or hopelessness and loss of interest in activities that were once enjoyed [35]. Antidepressant monotherapy was defined as patients taking a single antidepressant agent to treat a major depressive disorder. All respondents who were identified as having depression in the 2012–2019 MEPS database, were aged over 19 years, and taking a single agent of Sertraline, Fluoxetine or Escitalopram, were included in the study. To appreciate the effects of the medicine on changes in depressive symptoms during the study, only participants who started taking antidepressants at round 2 and round 3 of the panel were included in the study. The “purchrd” variable was used to select participants from various rounds of the panel. The rationale was to compare the baseline depressive symptoms of the participants from the time they started taking the medications with their symptoms after they had been taking them for roughly a year (in round 4). This will enable us gain insights into the effects of the medicine on the change in depressive symptoms during the study. Patients who purchased medicine before or at the beginning of rounds 1, 4, and 5 of the panel were excluded from the study. Patients who were taking combination therapy were excluded from the study. Patients who had comorbid conditions were also excluded from the study. Respondents with missing responses on the dependent variable (K6 scores) were excluded from the analysis.

### 2.2. Study Design

The MEPS HC medical condition file was used to identify individuals with depression. The MEPS medical condition file contains information on the observation of each self-reported medical condition that a respondent experienced during the data collection year. Medical conditions reported by participants were recorded by interviews and coded to fully specified ICD-10-CM and ICD-9-CM codes. Depression was identified using ICD-9-code 296, 311, and ICD-10-code F32 [34].

Patients taking antidepressants were identified using the prescribed medicines file (Figure 2). The most commonly used antidepressants, Fluoxetine, Escitalopram, and Sertraline, were identified using “rxname” and “rxdrgnam” variables from the prescribed medicines file [34].

The patients’ demographic characteristics were identified from the patient characteristic file. In this study, we included age, race, and gender.

### 2.3. Outcome Measures

The effect of the medicines on psychological distress was assessed using the Kessler Index (K6) scores. The Kessler Index (K6) scores measure individuals’ non-specific psychological distress in the past 30 days [28]. The scale consists of six items, each rated on a five-point Likert scale (from “none of the time” to “all of the time”) [28]. Appendix A.

The longitudinal data files in the MEPS contain K6 scores. These scores are measured in rounds 2 and 4 of a panel and are roughly a year apart [36]. Previously reported cut off-points in the literature were used to stratify K6 scores into no/low psychological distress (0–6), mild–moderate psychological distress (7–12), and severe distress (13–24) [28].

In this study, regarding changes in the baseline K6 score (that is round 2–round 4), 1–24 was identified as improved, whereas a change in the K6 score of 0 was classified as unchanged, and when a change in the baseline K6 score ranged from −1 to −24, it was classified as having declined.

### 2.4. Statistical Analysis

Descriptive statistics were used to describe the population according to their socio-demographic characteristics. All statistical values were considered significant at a level of significance of *p* ≤ 0.05. The dependent variable, namely the difference in K6 scores, was categorized using 1–24 as “improved”, −1–−24 as “declined” and 0 as “unchanged”. A multinomial logistic regression model was built to determine the effect of the independent variables on the above-mentioned dependent variable. Demographic variables such as race, gender, and age were controlled in the regression analysis. Statistical analysis was conducted using STATA software (version 15.1).

## 3. Results

### Demographic Characteristics of Study Population

Table 1 shows the demographic characteristics of the study population for each antidepressant. Among the three antidepressants used in the analysis, Sertraline was the most utilized medication among the study population (N = 251, 42.61%) followed by Fluoxetine (N = 185, 31.41%). Most of the study population were females (N = 417), representing 70.5% of the total study sample. Among different races, non-Hispanic whites were the highest users (N = 489, 83.02%) of the three SSRIs, with American Indians being the lowest users (N = 9, 1.53%) of the three SSRIs. Most of the study population was within the 40–59 age group (N = 244, 38.54%).

Table 2 shows the percentage of patients on Sertraline, Fluoxetine, and Escitalopram who showed improvement, no change, or decline in Kessler 6 scores. The majority of the patients (N = 467, 92.48%) were in the improved group, regardless of which of the three medications they were taking. Fluoxetine had the highest improvement rate of 94.27%, compared with Sertraline, which had an improvement rate of 91.96%, and Escitalopram, which had an improvement rate of 91.13%.

Table 3 shows the multinomial logistic regression results for changes in the Kessler Index scores among patients taking Sertraline, Fluoxetine, or Escitalopram monotherapy. A total of 84 participants with missing responses on the Kessler Index score were excluded, resulting in 505 participants being included in the regression analysis. Participants in the unchanged K6 category were used as references to predict improvement in psychological distress for users on the three SSRIs. Moreover, participants taking Fluoxetine were treated as the reference group among the three medications. Among the various age groups, participants aged between 20 and 39 years were used as the reference group, while non-Hispanic whites were used as the reference for race. In comparison with the participants taking Fluoxetine, the results did not show any statistical difference between participants taking Escitalopram (OR = 0.2823, 95% CI, 0.0209–3.812; *p* = 0.34) and those taking Sertraline (OR = 0.45, 95% CI, 0.06–3.3249; *p* = 0.43).

## 4. Discussion

Patients with a major depressive disorder usually have deteriorating mental health that affects the physical and social aspects of their lives. The primary aim of this study was to assess the effects of Sertraline, Fluoxetine, and Escitalopram on psychological distress using changes in Kessler Index 6 scores among adult outpatients diagnosed with one major depressive disorder.

The study sample was characterized by over 70% women, which corresponds with other studies that show that women are more likely than men to experience more depression. Females are also more likely than men to report to a mental health facility or seek medical attention [37,38]. In addition, the increased prevalence of depression correlates with hormonal changes in women, particularly during puberty, before menstruation, following pregnancy, and at perimenopause, suggesting that female hormonal fluctuations may trigger depression [39,40].

The majority of the study population were non-Hispanic whites. Similar racial/ethnic differences in antidepressant use are observed in the treatment of depression [41]. It has also been reported that factors such as racial/ethnic variation in mental health services and availability, treatment acceptability, and educational factors play a role in the prevalence of depression and antidepressant use among races [42]. The 40–59 age group was the highest population taking antidepressant monotherapy, representing over 38% of the study sample. On the contrary, recent studies have shown that young adults aged 18–29 have a higher prevalence of depression than older adults [43,44]. In part, the COVID-19 pandemic has been identified to have played a major role in the increase in the prevalence of depression among young adults [30,31,32]. Young adults have suffered from higher levels of depression and anxiety than older adults throughout the pandemic [45]. According to the Centers for Disease Control and Prevention’s (CDC) Household Pulse Survey, 36% of 18–29-year-olds had symptoms of depression in early May 2021, compared to 22% of those aged 40–49 and 15% of those aged 50–59 [45].

The descriptive statistics showed that 94.27% of the study participants taking Fluoxetine had experienced an improvement in their psychological distress after one year on the medication, followed by Sertraline (91.96%) and Escitalopram (91.13%). The overall improvement rate of 92.48% among the study sample indicates only that selective serotonin reuptake inhibitor medication effectively improves patient-reported outcomes, specifically psychological distress, over one year of taking the medication. In a similar study, the majority of the participants taking either first- or second-generation antidepressant monotherapy remained in the unchanged category after round 4 [36]. The authors explained that the medications might have elicited desirable responses resulting in patients having controlled depressive symptoms even at the time of the initial measure (round 2 of the panel) of psychological distress [37].

The current study compared the impact of Fluoxetine, Sertraline, and Escitalopram on patient-reported outcomes and psychological distress using changes in the Kessler 6 score. In our comparison with Fluoxetine as a reference drug, there was no statistical difference observed between the effect of Sertraline (OR = 0.45, 95% CI, 0.06–3.3249; *p* = 0.43), and Escitalopram (OR = 0.2823, 95% CI, 0.0209–3.812; *p* = 0.34) on psychological distress. Currently, there is insufficient data on evaluating the effectiveness of these commonly prescribed antidepressants using changes in the Kessler 6 score as a patient-reported outcome. A similar study on changes in the Kessler Index 6 score showed no significant difference between patients using monotherapy and those using add-on/switch therapy [36]. However, comparing our results to a meta-analysis involving 24,595 participants in 111 studies on the efficacy and acceptability of 12 antidepressants, Escitalopram, Sertraline, and Fluoxetine were found to have superior efficacy than the SNRIs in the meta-analysis [46]. With Fluoxetine as a reference compound, both Escitalopram and Sertraline had a significantly higher efficacy rate than Fluoxetine. However, they concluded that Sertraline may be preferable because of the balance between its efficacy and its tolerability [46,47]. In these studies, the treatment effect was measured using another instrument variable, changes in the baseline Montgomery–Asberg Depression Rating Scale (MADRS) total score.

The strength of this study was that a retrospective longitudinal database was used with a nationally representative sample. Due to the structure of the Medical Expenditure Panel Survey (MEPS), we were able to assess the outcome of the medications on psychological distress over time points approximately one year apart (from round 2 to round 4). This gives adequate time to elicit rich data on the long-term effect of the medications on the participants, which is essential for a chronic disease with a high relapse rate, such as depression. However, there were limitations to the study. This study focused on patients with a major depressive disorder without any comorbidities. This limits the generalizability of the results. The study is susceptible to response bias, as the information is self-reported by respondents and cannot therefore always be considered reliable. Moreover, this study could not adjust for the type and severity of depression, illness duration, side effects, and medication adherence, due to the structure of the MEPS. Additionally, this study could not account for the specific dose and titration of the medication, due to the nature of the MEPS, which does not provide dose-related information on the medications. We assumed that patients were prescribed the standard dose of the medications: Escitalopram 10–20 mg once a day [48], Sertraline 150–200 mg daily [49], and Fluoxetine 20–60 mg per day [50]. A future study could focus on examining the real-world impacts of these most widely prescribed antidepressants together with newly approved antidepressants, taking into account medication adherence, the tolerability of the medications, and the type and severity of depression. Due to insufficient evidence on the real-world impacts of selective serotonin reuptake inhibitors among depressed patients, this study adds to the evidence available to inform clinicians on the effect of the long-term use of selective serotonin reuptake inhibitors on patient-reported outcomes among patients with chronic depression. This study can also serve as a guide for researchers in this area, who can focus on the use of second-generation antidepressant monotherapy and dual-therapy antidepressants among patients with severe depression using real-world data.

## 5. Conclusions

Based on the descriptive statistics, all the medications effectively improve the rate of psychological distress among adult patients suffering from major depressive disorders without comorbid conditions. Moreover, no significant difference in the improvement rate of psychological distress for the participants was observed in our comparison of the three selective serotonin reuptake inhibitors. In addition to taking the effectiveness of the medications into account, it is imperative that clinicians consider patients’ preferences and tolerability toward specific antidepressant medications in their prescribing decisions.

## Figures and Tables

**Figure 1 healthcare-11-00740-f001:**
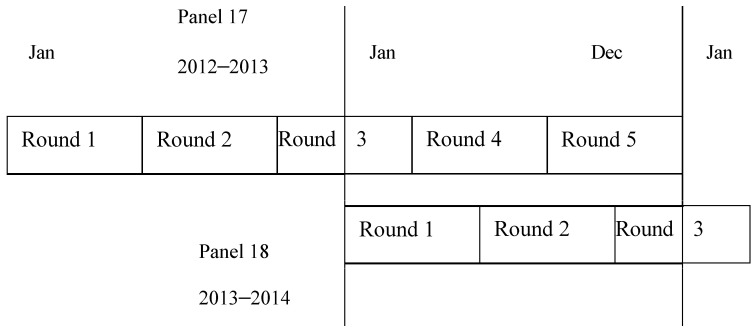
Adopted from MEPS AHRQ panel design [34].

**Figure 2 healthcare-11-00740-f002:**
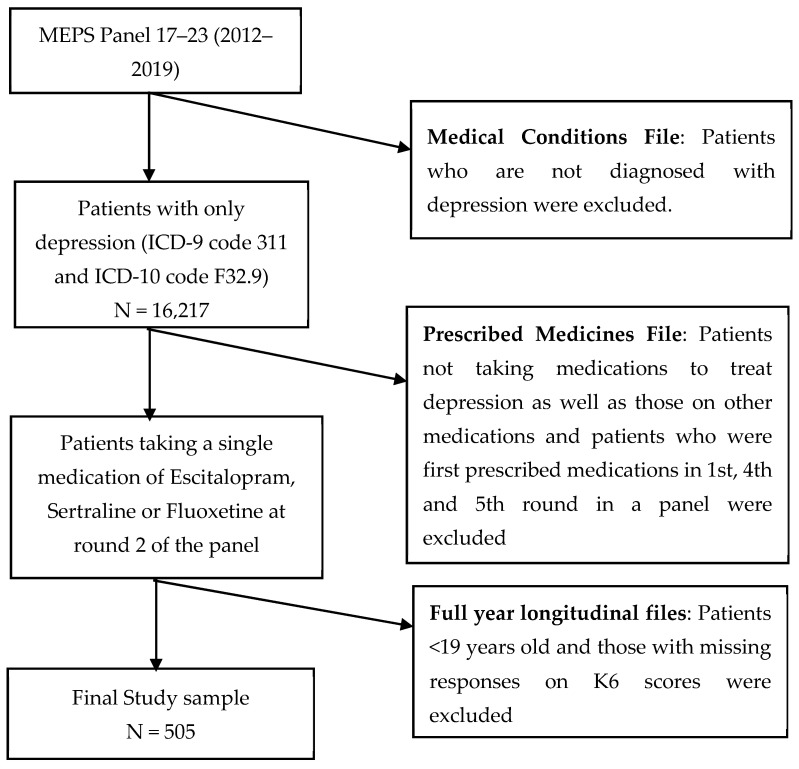
Flowchart of study cohort [28,34].

**Table 1 healthcare-11-00740-t001:** Demographic characteristics of study participants on the various antidepressants.

Characteristics	Escitalopram N (%)	Sertraline N (%)	Fluoxetine N (%)
Total	153 (25.98)	251 (42.61)	185 (31.41)
Gender			
Male	34 (22)	73 (29)	67 (36)
Female	119 (78)	178 (71)	118 (64)
Age			
20–39	48 (31.2)	80 (31.9)	54 (29.0)
40–59	65 (42.5)	99 (39.6)	80 (43.1)
60–80	40 (26.2)	72 (28.5)	51 (27.9)
Race			
White	129 (84.31)	205 (81.67)	155 (83.78)
Black	11 (7.19)	29 (11.55)	15 (8.11)
American Indian	1 (0.65)	5 (1.99)	3 (1.62)
Asian	4 (2.61)	4 (1.59)	7 (3.78)
Multi-race	8 (5.23)	8 (3.19)	5 (2.70)

**Table 2 healthcare-11-00740-t002:** Percentage of individuals showing a change in K6 scores based on Fluoxetine, Escitalopram, Sertraline.

Second-Generation Antidepressants	Improved N (%) 467 (92.48)	Unchanged N (%) 31 (6.14)	Declined N (%) 7 (1.39)
Fluoxetine	148 (94.27)	6 (3.82)	3 (1.91)
Escitalopram	113 (91.13)	10 (8.06)	1 (0.81)
Sertraline	206 (91.96)	15 (6.70)	3 (1.34)

**Table 3 healthcare-11-00740-t003:** Multinomial logistic regression to predict improvement in K6 scores among SSRIs users.

Category Ref: Unchanged	Improvement Rate		Declined Rate	
	OR (95% CI)	*p*-Value	OR (95% CI)	*p*-Value
Drug ref: Fluoxetine				
Escitalopram	0.2823 (0.0209–3.812)	0.340	0.4269 (0.1209–1.5067)	0.185
Sertraline	0.4500 (0.06–3.3249)	0.433	1.088 (0.2885–4.1027)	0.901
Sex: Female				
Male	0.839 (0.466–1.212)	0.530	0.605 (0.081–1.129)	0.15
Age ref: 20–39				
40–59	0.9911 (0.0998–9.839)	0.994	1.8426 (0.5442–6.2379)	0.325
60–80	0.6620 (0.0553–7.922)	0.744	0.9330 (0.2930–2.9701)	0.906
Race ref: White				
Black	33.304 (2.671–415.19)	0.007	8.7937 (1.099–70.3069)	0.040
American Indian	6.22 (0.3324–116.39)	0.221	1.50 (1.35–1.66)	0.00
Asian	5.31 (0.2243–125.79)	0.300	0.5961 (0.0688–5.1633)	0.638
Multi-race	1.80 (0.356–9.095)	0.476	4.43 (3.88–4.98)	0.00

## Data Availability

Publicly available datasets were analyzed in this study. This data can be found here: https://www.meps.ahrq.gov/mepsweb/ (accessed on 21 September 2021).

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
