# Peer review of "Impact of Sertraline, Fluoxetine, and Escitalopram on Psychological Distress among United States Adult Outpatients with a Major Depressive Disorder"

_healthcare, 2023, doi:10.3390/healthcare11050740_

Round 1

Reviewer 1 Report

I believe that the problem analyzed by the researchers is a valuable contribution to further research. Hence it deserves to be published.

1.                          2.1. Data source

The current retrospective longitudinal study was conducted to examine the effectiveness 129 of Sertraline, Fluoxetine, and Escitalopram monotherapy on psychological distress as a 130 patient-reported outcome using the Medical Expenditure Panel Survey (MEPS). Data 131 from MEPS used in this study, spanned from January 1, 2012, to December 31, 2019 (panel 132 17 to panel 23) [35]. (Can add where and among which national group) (128-133).

2. For example, in the correct manuscript, K6scores were excluded from the analysis (163).

3. Will check the bibliography

Reddy, M. (2010). Depression: The Disorder and the Burden. Indian Journal of Psychological Medicine, 32(1), 1–2. 369 https://doi.org/10.4103/0253-7176.70510

Reddy, M. S. (2010)………

Author Response

Dear reviewer,

Thank you for your comments. We have responded to your comments point by point as follows:

Comments and suggestions for authors

  1. 1. Data source

The current retrospective longitudinal study was conducted to examine the effectiveness 129 of Sertraline, Fluoxetine, and Escitalopram monotherapy on psychological distress as a 130 patient-reported outcome using the Medical Expenditure Panel Survey (MEPS). Data 131 from MEPS used in this study, spanned from January 1, 2012, to December 31, 2019 (panel 132 17 to panel 23) [35]. (Can add where and among which national group) (128-133).

Response: We have included the national group “non-institutionalized U.S. civilians” in the statement in line 170.

  1. For example, in the correct manuscript,K6scores were excluded from the analysis (163).

Addressed in line …

Response: We excluded respondents with missing responses on our dependent variable which was the K-6 score from our multinomial regression analysis. The wording of our statement was confusing. In the sentence “and” should not have been there. This has been addressed in lines 250-251.

  1. Will check the bibliography

Reddy, M. (2010). Depression: The Disorder and the Burden. Indian Journal of Psychological Medicine, 32(1), 1–2. 369 https://doi.org/10.4103/0253-7176.70510

Reddy, M. S. (2010)………

Response: We have updated the first statement of our introduction with current information that gives an overview of the burden of depression on healthcare system. This has been addressed in line 31-32.

Reviewer 2 Report

Kind Authors,

here are some small suggestions for your paper:

- the paper is well-written;
- the introduction is slightly verbose; it might be helpful to paraphrase some parts, especially from line 45 to line 66 and from line 94 to line 117;
- the Kessler Index (K6) presented from line 191 to line 198 could be included in the Supplementary Materials to improve the readability of the paper;
- sex was not included in the multinomial logistic regression and I would ask you, if possible, to include it (or specify the reason for exclusion);
- in the discussion of limitations, I suggest to include the fact that there is no references about the dosage of SSRIs taken or prescribed by the patient; this could be a limitation due to dosage may affect remission of depression.

Author Response

Dear reviewer,

Thank you for your valuable comments. We have responded to the comments points by points as follow:

- the paper is well-written.

Response: Thank you very much.

  1. - the introduction is slightly verbose; it might be helpful to paraphrase some parts, especially from line 45 to line 66 and from line 94 to line 117;

Response: We have addressed verbosity from line 45 to 66 and line 94 to line 117. These changes could be seen in the updated version from lines 50-64 and lines 92-156 respectively.

  1. - the Kessler Index (K6) presented from line 191 to line 198 could be included in the Supplementary Materials to improve the readability of the paper;

Response: We have included Kessler Index (K6) as part of the supplementary labelled as supplementary 1 on line 291.

  1. - sex was not included in the multinomial logistic regression and I would ask you, if possible, to include it (or specify the reason for exclusion);

Response: That was an oversight in our draft. Sex has been added to the multinomial logistic regression analysis. This has been addressed in the table 3 between line 368 and 369

  1. - in the discussion of limitations, I suggest to include the fact that there is no references about the dosage of SSRIs taken or prescribed by the patient; this could be a limitation due to dosage may affect remission of depression.

Response: This has been addressed from line 461-465.

Reviewer 3 Report

It was my pleasure to review the manuscript “Impact of Sertraline, Fluoxetine, and Escitalopram on psychological distress among United States adult outpatients with Major Depressive Disorder” The article contains an interesting summary of findings on the comparative effectiveness among three, commonly used antidepressants on psychological distress as a  patient-reported outcome.

The abstract is well-written and concise.

The introduction section is well-written and concise.

 Material and methods

“MEPS is a nationally representative estimates of health care use, expenditures, sources of payment, health insurance coverage, demographic characteristics, respondents' health status, employment, access to care and satisfaction with healthcare” at line 167 is repeating line 134.

The dosages of medications were not discussed as well as the titration schedule. Did patients remain on the same dosages all the time?

The side effects profile did not discuss.

 If patients did not have psychiatric comorbidities, did they have medical comorbidities?

The results are well-presented and instructive.

The discussion is well written, but the significance of the study did not reflect the lack of new knowledge for practitioners.

Limitations of the study addressed. 

Author Response

Dear reviewer,

Thank you for your valuable comments. We have responded to the comments points by points as follow:

  1. The abstract is well-written and concise.

Response: Thank you.

  1. The introduction section is well-written and concise.

Response: Thank you.

 Material and methods

  1. “MEPS is a nationally representative estimates of health care use, expenditures, sources of payment, health insurance coverage, demographic characteristics, respondents' health status, employment, access to care and satisfaction with healthcare” at line 167 is repeating line 134.

Response: This was an oversight in our draft.  Line 167 in the previous version had been deleted from the updated version of the manuscript.

  1. The dosages of medications were not discussed as well as the titration schedule. Did patients remain on the same dosages all the time?

Response: Due to the nature of the datasets used from MEPS, information on exact dose and titration of the medication as well as side effects are not provided therefore, we could not account for them. This has been addressed in our limitation line 461-465, where we assume patients were prescribed standard dose of the medications.

  1. The side effects profile did not discuss.

Response: Due to the nature of MEPS, which did not provide information on side effects, we could not account for the presence of side effects in the participants on the medications. This was addressed as a limitation in our study in line 460.

  1. If patients did not have psychiatric comorbidities, did they have medical comorbidities?

Response: The study focused on participants taking only antidepressants as long-term medication hence participants selected were those without any other comorbid condition which limits our generalizability as stated in line 457.

  1. The results are well-presented and instructive.

Response: Thank you

  1. The discussion is well written, but the significance of the study did not reflect the lack of new knowledge for practitioners.

Response: This has been addressed in lines 468-472.  

  1. Limitations of the study addressed. 

Response: Thank you

Round 2

Reviewer 3 Report

Thanks for responding to my comments.